# A New Estimator for Standard Errors with Few Unbalanced Clusters

**Gianmaria Niccodemi [1,*] and Tom Wansbeek [2]**

1   Faculty of Humanities, Education and Social Sciences, University of Luxembourg, Esch-sur-Alzette, L-4366 Luxembourg, Luxembourg
2   Faculty of Economics and Business, University of Groningen, 9700 AV Groningen, The Netherlands; t.j.wansbeek@rug.nl
*   Correspondence: gianmaria.niccodemi@uni.lu

**Abstract:** In linear regression analysis, the estimator of the variance of the estimator of the regression coefficients should take into account the clustered nature of the data, if present, since using the standard textbook formula will in that case lead to a severe downward bias in the standard errors. This idea of a cluster-robust variance estimator (CRVE) generalizes to clusters the classical heteroskedasticity-robust estimator. Its justification is asymptotic in the number of clusters. Although an improvement, a considerable bias could remain when the number of clusters is low, the more so when regressors are correlated within cluster. In order to address these issues, two improved methods were proposed; one method, which we call CR2VE, was based on biased reduced linearization, while the other, CR3VE, can be seen as a jackknife estimator. The latter is unbiased under very strict conditions, in particular equal cluster size. To relax this condition, we introduce in this paper CR3VE-$\lambda$, a generalization of CR3VE where the cluster size is allowed to vary freely between clusters. We illustrate the performance of CR3VE-$\lambda$ through simulations and we show that, especially when cluster sizes vary widely, it can outperform the other commonly used estimators.

**Keywords:** clustered data; few clusters; unbalanced clusters; cluster-robust variance estimator; inference

## 1. Introduction

In linear regressions with clustered data, it is common practice to estimate the variance of the estimated parameters using the cluster-robust variance estimator (CRVE from hereon) introduced by Liang and Zeger (1986), as a generalization of the White (1980) heteroskedastic-robust estimator. The justification is asymptotic, with number of clusters tending to infinity. Bell and McCaffrey (2002) show that in a finite context, with few clusters and error terms that are correlated within cluster, CRVE leads to severely downward-biased standard errors and thus to misleading inference about the estimated parameters. Moulton (1986, 1990) and Cameron and Miller (2015) point out that this issue is particularly relevant for regressors that are correlated within cluster such as policy variables that are implemented only in certain regions or states. An additional issue for inference about the estimated parameters is that, under the null hypothesis and with few clusters, the distribution of the test statistic is unknown and approximate normality cannot be claimed.

Following Bell and McCaffrey (2002), inferences about the estimated parameters can be improved by (i) reducing the bias of CRVE with either BRL (bias reduced linearization), also known as CR2VE, or the jackknife estimator $v_{JK}$, also known as CR3VE, both based on transformed OLS residuals; CR2VE and CR3VE generalize, using clustered data, the heteroskedasticity-consistent covariance estimators HC2 and HC3, introduced by MacKinnon and White (1985). Inference about the estimated parameters can be also improved by (ii) approximating the distribution of the test statistic with the *t*-distribution with an extension of the Satterthwaite (1946) degrees of freedom (DOF) that are data-determined

and regressor-specific. Imbens and Kolesar (2016) developed a more refined version of the data-determined regressor-specific DOF used by Bell and McCaffrey (2002).

Bell and McCaffrey (2002) also show that CR3VE tends to overestimate the standard errors. In this paper, we introduce CR3VE-$\lambda$, a cluster-robust variance estimator that is identical to CR3VE in the case of balanced clusters but, in the case of unbalanced clusters, takes the difference in cluster sizes into account such that the computed standard errors are less conservative and unbiased under more general conditions.

The paper is organized as follows. In Section 2, we discuss basic theory on CRVE, CR2VE and CR3VE. In Section 3, we introduce CR3VE-$\lambda$. In Section 4, we illustrate and test the performance of CRVE, CR2VE, CR3VE and CR3VE-$\lambda$ to compute standard errors with few clusters using Monte Carlo simulations. In Section 5, we present ideas for future research related to the current paper. Section 6 concludes the paper.

## 2. Basic Theory: CRVE, CR2VE and CR3VE

Consider the regression model $\mathbf{y} = \mathbf{X}\boldsymbol{\beta} + \boldsymbol{\varepsilon}$ with observations that can be grouped into $C$ clusters of size $n_1, \ldots, n_C$; $\sum_c n_c = n$. Write, for the $c$-th cluster, $\mathbf{y}_c = \mathbf{X}_c\boldsymbol{\beta} + \boldsymbol{\varepsilon}_c$, with $\mathrm{E}(\boldsymbol{\varepsilon}_c) = \mathbf{0}$ and $\mathrm{var}(\boldsymbol{\varepsilon}_c) = \mathbf{V}_c$. The $\mathbf{V}_c$'s are collected in the block-diagonal matrix $\mathbf{V}$. After OLS we have

$$\mathrm{var}(\hat{\boldsymbol{\beta}}) = (\mathbf{X}'\mathbf{X})^{-1}\mathbf{X}'\mathbf{V}\mathbf{X}(\mathbf{X}'\mathbf{X})^{-1} = (\mathbf{X}'\mathbf{X})^{-1}\left(\sum_c \mathbf{X}_c'\mathbf{V}_c\mathbf{X}_c\right)(\mathbf{X}'\mathbf{X})^{-1}. \tag{1}$$

An intuitively appealing cluster-robust variance estimator (CRVE) based on OLS residuals per cluster $\hat{\boldsymbol{\varepsilon}}_c$ is

$$\widehat{\mathrm{var}}(\hat{\boldsymbol{\beta}}) = (\mathbf{X}'\mathbf{X})^{-1}\left(\sum_c \mathbf{X}_c'\hat{\boldsymbol{\varepsilon}}_c\hat{\boldsymbol{\varepsilon}}_c'\mathbf{X}_c\right)(\mathbf{X}'\mathbf{X})^{-1}. \tag{2}$$

This estimator, which directly generalizes White (1980) and was introduced by Liang and Zeger (1986), is consistent when the number of clusters goes to infinity. The same holds when (2) is scaled, as in Stata, by the factor $C(n-1)/(C-1)(n-k)$, with $k$ the number of regressors. Since this factor is larger than one, it increases the estimated variance. In the case of few clusters, asymptotics will be a poor guide. In what follows, we therefore consider its bias instead.

Let $\mathbf{M} = \mathbf{I}_n - \mathbf{X}(\mathbf{X}'\mathbf{X})^{-1}\mathbf{X}'$, let $\mathbf{S}_c$ be the $n \times n_c$ matrix that selects the columns of $\mathbf{M}$ corresponding to cluster $c$, let $\mathbf{L}_c \equiv \mathbf{M}\mathbf{S}_c$ and let

$$\mathbf{H}_c \equiv \mathbf{S}_c'\mathbf{M}\mathbf{S}_c = \mathbf{I}_{n_c} - \mathbf{X}_c(\mathbf{X}'\mathbf{X})^{-1}\mathbf{X}_c'.$$

There holds $\mathbf{H}_c = \mathbf{L}_c'\mathbf{L}_c$ since $\mathbf{M}$ is idempotent and symmetric. With $\hat{\boldsymbol{\varepsilon}} = \mathbf{M}\boldsymbol{\varepsilon}$ and $\hat{\boldsymbol{\varepsilon}}_c = \mathbf{L}_c'\boldsymbol{\varepsilon}$, we then have $\mathrm{E}(\hat{\boldsymbol{\varepsilon}}_c\hat{\boldsymbol{\varepsilon}}_c') = \mathbf{L}_c'\mathbf{V}\mathbf{L}_c \neq \mathbf{V}_c$, so that

$$\mathrm{E}[\widehat{\mathrm{var}}(\hat{\boldsymbol{\beta}})] = (\mathbf{X}'\mathbf{X})^{-1}\left(\sum_c \mathbf{X}_c'\mathbf{L}_c'\mathbf{V}\mathbf{L}_c\mathbf{X}_c\right)(\mathbf{X}'\mathbf{X})^{-1} \neq \mathrm{var}(\hat{\boldsymbol{\beta}}). \tag{3}$$

To reduce the bias, consider choosing a variance estimator based on transformed residuals $\tilde{\boldsymbol{\varepsilon}}_c \equiv \mathbf{A}_c\hat{\boldsymbol{\varepsilon}}_c$, for some $\mathbf{A}_c$. Then

$$\mathrm{E}[\widehat{\mathrm{var}}(\hat{\boldsymbol{\beta}})] = (\mathbf{X}'\mathbf{X})^{-1}\left(\sum_c \mathbf{X}_c'\mathbf{A}_c\mathbf{L}_c'\mathbf{V}\mathbf{L}_c\mathbf{A}_c'\mathbf{X}_c\right)(\mathbf{X}'\mathbf{X})^{-1}.$$

From (1), unbiasedness requires the $\mathbf{A}_c$ to be such that $\mathbf{A}_c\mathbf{L}_c'\mathbf{V}\mathbf{L}_c\mathbf{A}_c' = \mathbf{V}_c$ for all $c$ uniformly in the $\mathbf{V}_c$. This is infeasible and therefore we consider two second-best solutions.

The first solution is to consider the case of no cluster effects, $\mathbf{V}_c = \sigma^2 \mathbf{I}_{n_c}$ for all $c$, and make the estimator unbiased for this case. Then $\mathrm{E}(\hat{\varepsilon}_c \hat{\varepsilon}_c') = \mathbf{L}_c' \mathbf{V} \mathbf{L}_c = \sigma^2 \mathbf{L}_c' \mathbf{L}_c = \sigma^2 \mathbf{H}_c$ and consequently

$$\mathrm{E}[\widehat{\mathrm{var}}(\hat{\boldsymbol{\beta}})] = \sigma^2 (\mathbf{X}'\mathbf{X})^{-1} \left( \sum_c \mathbf{X}_c' \mathbf{A}_c \mathbf{H}_c \mathbf{A}_c' \mathbf{X}_c \right) (\mathbf{X}'\mathbf{X})^{-1}. \tag{4}$$

The variance estimator is unbiased if $\mathbf{A}_c \mathbf{H}_c \mathbf{A}_c' = \mathbf{I}_{n_c}$ and so we choose $\mathbf{A}_c = \mathbf{H}_c^{-1/2}$. This estimator, introduced by Bell and McCaffrey (2002) and called BRL, is extensively discussed by Cameron and Miller (2015) and it is also known as CR2VE.

The second solution is based on the idea that the elements in $\mathbf{M}$ outside the blocks on the diagonal may be small. Then $\mathbf{L}_c$ can be approximated by a matrix with $\mathbf{H}_c$ as its $c$-th block and zeros outside this block. Then $\mathbf{L}_c' \mathbf{V} \mathbf{L}_c = \mathbf{H}_c \mathbf{V}_c \mathbf{H}_c$ and choosing $\mathbf{A}_c = \mathbf{H}_c^{-1}$ leads, when scaled by a factor $(C-1)/C$, to an estimator that is approximately unbiased when there are no cluster effects. This estimator with the jackknife correction is also introduced by Bell and McCaffrey (2002), who called it $v_{JK}$, it is discussed by Cameron and Miller (2015) and it is also known as CR3VE. CR2VE and CR3VE can be computationally intensive because they require the inversion of matrices of order equal to the cluster sizes. CR2VE and CR3VE can be computed efficiently, that is, with computing time and storage of order $O(n_c)$; a succinct proof is given by Niccodemi et al. (2020).

Both CR2VE and CR3VE are used in the literature as an alternative to bootstrapping. The bootstrap literature has evolved rapidly since Cameron et al. (2008) proposed the use of a wild cluster bootstrap procedure to improve inference in the case of few clusters. Generally, the wild cluster bootstrap procedure performs well. However, MacKinnon and Webb (2017) show that inference based on this procedure can fail in the case of dummy regressors equal to zero or one in very few clusters. Djogbenou et al. (2019) propose an asymptotic analysis of cluster-robust inference mainly focused on the wild cluster bootstrap procedure, proving its asymptotic validity under certain conditions on the cluster sizes. They show, both theoretically and through some experiments, how variation in cluster sizes affects the asymptotic validity of this procedure and they conclude that the wild cluster restricted bootstrap using the Rademacher distribution performs better than any other competitors.

## 3. From CR3VE to CR3VE-$\lambda$

To analyze the bias of CR3VE we scale (4) by $(C-1)/C$ and use

$$\mathbf{A}_c \mathbf{H}_c \mathbf{A}_c = \mathbf{H}_c^{-1} = \mathbf{I}_{n_c} + \mathbf{X}_c (\mathbf{X}'\mathbf{X} - \mathbf{X}_c' \mathbf{X}_c)^{-1} \mathbf{X}_c'$$

to obtain

$$\mathrm{E}[\widehat{\mathrm{var}}(\hat{\boldsymbol{\beta}})] = \frac{C-1}{C} \sigma^2 \left( (\mathbf{X}'\mathbf{X})^{-1} + \sum_c (\mathbf{X}'\mathbf{X})^{-1} \mathbf{X}_c' \mathbf{X}_c (\mathbf{X}'\mathbf{X} - \mathbf{X}_c' \mathbf{X}_c)^{-1} \mathbf{X}_c' \mathbf{X}_c (\mathbf{X}'\mathbf{X})^{-1} \right). \tag{5}$$

When clusters are balanced and have the same covariance structure then $\mathbf{X}_c' \mathbf{X}_c = \mathbf{X}'\mathbf{X}/C$ for all $c$, and (5) reduces to $\mathrm{E}[\widehat{\mathrm{var}}(\hat{\boldsymbol{\beta}})] = \sigma^2 (\mathbf{X}'\mathbf{X})^{-1}$. Thus, in the case of balanced clusters, CR3VE with the correction factor $(C-1)/C$ is unbiased.

We propose a different scaling factor than $(C-1)/C$ for CR3VE in the more general case of unbalanced clusters that still have the same covariance structure. Define $\pi_c \equiv n_c/n$ for cluster $c$. Then $\mathbf{X}_c' \mathbf{X}_c = \pi_c \mathbf{X}'\mathbf{X}$ and the expression in parentheses in (5) becomes $\lambda (\mathbf{X}'\mathbf{X})^{-1}$, with

$$\lambda \equiv 1 + \sum_c \frac{\pi_c^2}{1 - \pi_c},$$

and $\lambda \geq C/(C-1)$, with equality holding in the case of balanced clusters. To see this, let $\boldsymbol{\pi} \equiv (\pi_1, \ldots, \pi_C)'$, $\boldsymbol{\Pi} \equiv \mathrm{diag}(\boldsymbol{\pi})$, $\mathbf{a} \equiv (\mathbf{I}_C - \boldsymbol{\Pi})^{-1/2} \boldsymbol{\pi}$ and $\mathbf{b} \equiv (\mathbf{I}_C - \boldsymbol{\Pi})^{1/2} \boldsymbol{\iota}_C$,

$\mathbf{a}'\mathbf{a} = \boldsymbol{\pi}'(\mathbf{I}_C - \boldsymbol{\Pi})^{-1}\boldsymbol{\pi}, \mathbf{b}'\mathbf{b} = \boldsymbol{\iota}'_C(\mathbf{I}_C - \boldsymbol{\Pi})\boldsymbol{\iota}_C$, and $\mathbf{a}'\mathbf{b} = 1$. Since $(\mathbf{a}'\mathbf{b})^2 \leq \mathbf{a}'\mathbf{a}\,\mathbf{b}'\mathbf{b}$ there holds

$$\sum_c \frac{\pi_c^2}{1 - \pi_c} = \boldsymbol{\pi}'(\mathbf{I}_C - \boldsymbol{\Pi})^{-1}\boldsymbol{\pi} \geq \frac{1}{\boldsymbol{\iota}'_C(\mathbf{I}_C - \boldsymbol{\Pi})\boldsymbol{\iota}_C} = \frac{1}{C - 1},$$

so $\lambda - 1 \geq 1/(C-1)$ or $\lambda \geq C/(C-1)$. This suggests that $1/\lambda$ may be a better scaling factor than $(C-1)/C$. As $1/\lambda \leq (C-1)/C$, we propose a lower estimate of the variance than with CR3VE. This fits in well with the observation by Bell and McCaffrey (2002), as mentioned in the Introduction, that CR3VE tends to overestimate the standard errors. We denote this estimator, which is unbiased under more general conditions than CR3VE, by CR3VE-$\lambda$.

## 4. Monte Carlo Simulations

We run several sets of Monte Carlo (MC) simulations and compare the bias of the standard errors based on unclustered standard errors (UN), CRVE, CR2VE and CR3VE with the bias of the standard errors based on CR3VE-$\lambda$. In each simulation, we generate randomly $C$ unbalanced clusters with number of observations per cluster $n_c \sim U\{1000 - g, 1000 + g\}$, where $g$ is different in each set of simulations. In other words, $n_c$ is drawn from a uniform distribution with constant mean but standard deviation that depends on $g$. We generate our dependent variable $y_{hc} = \alpha + \beta x_{hc} + \gamma d_c + e_{hc}$, where $h$ identifies the single observation (e.g., household) and $c$ identifies the $C$ clusters of size $n_c = n_1, \ldots, n_C$, and where $x_{hc} = q_{hc} + z_c$ and $e_{hc} = w_{hc} + u_c$. Moreover, $q_{hc}, z_c, w_{hc}, u_c$ are independently drawn from $N(0,1)$, $\alpha = 0$ and $\beta = \gamma = 1$, and $d_c$ is a dummy variable constant within cluster and randomly constrained, in each simulation, to be equal to 1 in half of the randomly generated clusters. The simulation set-up is somewhat similar to the one in Cameron et al. (2008). As pointed out by Cameron and Miller (2015), unclustered standard errors and CRVE are likely to be severely biased if the cluster effect and the correlation of the regressors within cluster are different from zero. Therefore, we set up experiments that allow both $e_{hc}$ and the regressors to be correlated within cluster, including the extreme case of $d_c$, a dummy variable that is constant within cluster. The presence of regressors correlated within cluster implies that the assumption under which CR3VE and CR3VE-$\lambda$ are unbiased are not met. Yet, CR3VE-$\lambda$ takes into account the difference in cluster size and, as this difference increases, it is expected to be less biased than CR3VE.

We run 100,000 simulations for each MC set and each MC set differs with respect to the number of clusters $C$ and $g$. We show results for $C = 4$ and $C = 6$, and for $g = 0$ (i.e., balanced clusters), $g = 250$, $g = 500$, $g = 900$ and $g = 990$, with standard deviation of the cluster size equal to 0, 145, 289, 520 and 572, respectively. For each simulation: (i) we compute the true standard deviation of $\hat{\boldsymbol{\beta}}$, sd($\hat{\boldsymbol{\beta}}$), based on

$$\text{var}(\hat{\boldsymbol{\beta}}) = (\mathbf{X}'\mathbf{X})^{-1}\left(\sum_c \mathbf{X}'_c \mathbf{V}_c \mathbf{X}_c\right)(\mathbf{X}'\mathbf{X})^{-1},$$

where

$$\mathbf{V}_c = \mathbf{I}_{n_c} + \boldsymbol{\iota}_c\boldsymbol{\iota}'_c,$$

and where $\boldsymbol{\beta} = (\alpha, \beta, \gamma)$, (ii) we compute the standard errors of $\hat{\boldsymbol{\beta}}$ and of $\hat{\gamma}$ based on the different methods $\text{se}_{UN}$, $\text{se}_{CRVE}$, $\text{se}_{CR2VE}$, $\text{se}_{CR3VE}$ and $\text{se}_{CR3VE-\lambda}$, (iii) we compute the difference between the standard errors based on the different methods and the true standard deviations sd($\hat{\boldsymbol{\beta}}$) and sd($\hat{\gamma}$). Finally, for each MC set we compute the mean of this difference (i.e., the estimated bias) for each method to compute the standard errors. From Tables 1 and 2 we can see that CR3VE-$\lambda$ always leads to the least biased standard errors, with estimated bias always close to zero. Moreover, it remarkably reduces the estimated bias of CR3VE with high unbalancedness. This is especially true for the dummy variable $d_i$.

**Table 1.** Estimated bias of $se(\hat{\beta})$ based on different methods: 100,000 Monte Carlo simulations.

|  | Balanced | Std. Deviation Cluster Size | | | |
|---|---|---|---|---|---|
|  |  | 145 | 289 | 520 | 572 |
| **4 clusters** | | | | | |
| $\hat{\mathrm{E}}[\mathrm{sd}(\hat{\beta})]$ | 0.1978 | 0.1967 | 0.1929 | 0.1790 | 0.1745 |
| $\widehat{\mathrm{Bias}}[se_{UN}(\hat{\beta})]$ | −0.1820 | −0.1809 | −0.1769 | −0.1628 | −0.1581 |
| $\widehat{\mathrm{Bias}}[se_{CRVE}(\hat{\beta})]$ | −0.1293 | −0.1271 | −0.1207 | −0.1069 | −0.1043 |
| $\widehat{\mathrm{Bias}}[se_{CR2VE}(\hat{\beta})]$ | −0.0667 | −0.0663 | −0.0644 | −0.0605 | −0.0599 |
| $\widehat{\mathrm{Bias}}[se_{CR3VE}(\hat{\beta})]$ | 0.0191 | 0.0192 | 0.0188 | 0.0164 | 0.0157 |
| $\widehat{\mathrm{Bias}}[se_{CR3VE-\lambda}(\hat{\beta})]$ | 0.0191 | 0.0184 | 0.0157 | 0.0066 | 0.0040 |
| **6 clusters** | | | | | |
| $\hat{\mathrm{E}}[\mathrm{sd}(\hat{\beta})]$ | 0.1839 | 0.1837 | 0.1829 | 0.1811 | 0.1807 |
| $\widehat{\mathrm{Bias}}[se_{UN}(\hat{\beta})]$ | −0.1709 | −0.1707 | −0.1699 | −0.1679 | −0.1675 |
| $\widehat{\mathrm{Bias}}[se_{CRVE}(\hat{\beta})]$ | −0.0775 | −0.0774 | −0.0792 | −0.0844 | −0.0868 |
| $\widehat{\mathrm{Bias}}[se_{CR2VE}(\hat{\beta})]$ | −0.0301 | −0.0300 | −0.0325 | −0.0386 | −0.0413 |
| $\widehat{\mathrm{Bias}}[se_{CR3VE}(\hat{\beta})]$ | 0.0198 | 0.0208 | 0.0199 | 0.0202 | 0.0195 |
| $\widehat{\mathrm{Bias}}[se_{CR3VE-\lambda}(\hat{\beta})]$ | 0.0198 | 0.0204 | 0.0182 | 0.0142 | 0.0120 |

**Table 2.** Estimated bias of $se(\hat{\gamma})$ based on different methods: 100,000 Monte Carlo simulations.

|  | Balanced | Std. Deviation Cluster Size | | | |
|---|---|---|---|---|---|
|  |  | 145 | 289 | 520 | 572 |
| **4 clusters** | | | | | |
| $\hat{\mathrm{E}}[\mathrm{sd}(\hat{\gamma})]$ | 1.0209 | 1.0250 | 1.0369 | 1.0847 | 1.1066 |
| $\widehat{\mathrm{Bias}}[se_{UN}(\hat{\gamma})]$ | −0.9805 | −0.9843 | −0.9957 | −1.0416 | −1.0623 |
| $\widehat{\mathrm{Bias}}[se_{CRVE}(\hat{\gamma})]$ | −0.4700 | −0.4790 | −0.5038 | −0.6066 | −0.6533 |
| $\widehat{\mathrm{Bias}}[se_{CR2VE}(\hat{\gamma})]$ | −0.1868 | −0.1953 | −0.2181 | −0.3191 | −0.3703 |
| $\widehat{\mathrm{Bias}}[se_{CR3VE}(\hat{\gamma})]$ | 0.1005 | 0.1000 | 0.1023 | 0.1054 | 0.1068 |
| $\widehat{\mathrm{Bias}}[se_{CR3VE-\lambda}(\hat{\gamma})]$ | 0.1005 | 0.0960 | 0.0856 | 0.0410 | 0.0225 |
| **6 clusters** | | | | | |
| $\hat{\mathrm{E}}[\mathrm{sd}(\hat{\gamma})]$ | 0.8306 | 0.8355 | 0.8506 | 0.9059 | 0.9276 |
| $\widehat{\mathrm{Bias}}[se_{UN}(\hat{\gamma})]$ | −0.7965 | −0.8013 | −0.8163 | −0.8706 | −0.8919 |
| $\widehat{\mathrm{Bias}}[se_{CRVE}(\hat{\gamma})]$ | −0.2478 | −0.2531 | −0.2786 | −0.3628 | −0.3953 |
| $\widehat{\mathrm{Bias}}[se_{CR2VE}(\hat{\gamma})]$ | −0.0837 | −0.0861 | −0.1057 | −0.1653 | −0.1894 |
| $\widehat{\mathrm{Bias}}[se_{CR3VE}(\hat{\gamma})]$ | 0.0524 | 0.0556 | 0.0514 | 0.0564 | 0.0610 |
| $\widehat{\mathrm{Bias}}[se_{CR3VE-\lambda}(\hat{\gamma})]$ | 0.0524 | 0.0537 | 0.0436 | 0.0265 | 0.0223 |

We acknowledge that the reader might be particularly interested in comparing the inferential performance of the various CRVEs, including CR3VE-$\lambda$, especially in a real-data setting. For this purpose we refer the reader to Niccodemi et al. (2020), where inferential results based on the Current Population Survey data clustered in few, highly unbalanced clusters and the *t*-distribution using the Imbens and Kolesar (2016) DOF are reported. This experiment is similar to the one developed by Cameron and Miller (2015), although more focused on cluster unbalancedness. According to the results, with few, highly unbalanced clusters CR3VE-$\lambda$ appears to be among the most promising methods for inference, as CR3VE tends to underreject a true null hypothesis.

## 5. A Note on Future Research

Future research on cluster-robust variance estimators, directly linked to the current work, might take at least two directions. First, Djogbenou et al. (2019) show through some experimental designs how the variation in cluster sizes affects the asymptotic validity

of the wild cluster bootstrap. Testing how CR3VE-$\lambda$ performs, in comparison to CR2VE and CR3VE and using the same experimental designs, might provide further elements to evaluate its performance.

Second, the effective number of clusters introduced by Carter et al. (2017) might be of particular interest for CR3VE-$\lambda$. The effective number of clusters depends, among others, on the cluster sizes. If the effective and the nominal number of clusters differ remarkably, and if this difference is, to some extent, due to heterogeneity in cluster sizes, then inference using CR3VE-$\lambda$ might be much more accurate then inference based on CR3VE. Therefore, it would be interesting to develop experiments that focus on the interaction between the effective number of clusters as a diagnostic tool and the use of CR3VE-$\lambda$ instead of CR3VE for inference. Of course, other possibilities include the use of the effective number of clusters to construct the scaling factor for CR3VE and the introduction of measures of the effective size of the clusters to compute CR3VE-$\lambda$.

## 6. Conclusions

We propose CR3VE-$\lambda$, an estimator for clustered standard errors that improves the jackknife estimator and is unbiased under more general conditions in the case of few unbalanced clusters. In simulations, CR3VE-$\lambda$ reduces the bias of CR3VE as the unbalancedness of the clusters increases. We also provide a reference to a longer working paper (i.e., Niccodemi et al. (2020)) that develops simulation results to compare inference based on CRVE, CR2VE, CR3VE and CR3VE-$\lambda$. Given the results of both sets of simulations, we suggest researchers to prefer CR3VE-$\lambda$ to CR3VE in the case of (few) highly unbalanced clusters.

For all the computations and the empirical illustrations we used Stata/SE 15.0. This paper comes with a Stata do-file that can be used with any cross-sectional dataset for the efficient computation of the standard errors based on CRVE, CR2VE, CR3VE and CR3VE-$\lambda$ and with a Stata do-file to replicate the Monte Carlo simulations. The Stata do-files are available upon request.

**Author Contributions:** All authors have contributed equally to the research. All authors have read and agreed to the published version of the manuscript.

**Funding:** This research received no external funding.

**Institutional Review Board Statement:** Not applicable.

**Informed Consent Statement:** Not applicable.

**Data Availability Statement:** Not applicable.

**Acknowledgments:** We are grateful to Viola Angelini, Rob Alessie, Nick Koning, Erik Meijer, Douglas Miller, Ulrich Schneider, Roberto Wessels and four referees for their helpful comments and suggestions.

**Conflicts of Interest:** The authors declare no conflict of interest.

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
