# Peer review of "A New Estimator for Standard Errors with Few Unbalanced Clusters"

_econometrics, doi:10.3390/econometrics10010006_

Round 1

Reviewer 1 Report

See my report.

Author Response

The attached PDF "Reply to referees" addresses the points raised by all 4 referees

Reviewer 2 Report

  •  Moderate English changes required
  • The manuscript should be formatted according to the Econometrics template
  • The reviewer suggests implementing the background part of the abstract within the 200 words limit

Author Response

(The authors gave the same response as above.)

Reviewer 3 Report

In this paper, a new type of cluster robust variance estimator (CRVE), that takes into account the possible difference in groups' size, is proposed. Moreover, it is unbiased under more general conditions than the previous CR3VE. The paper contribution is clear and adds new evidence to the literature. I am happy to say that I enjoyed the reading. However, I have some remarks. To summarize, the authors should increase their effort in improving the presentation of previous literature, especially concerning the problem of unequally sized clusters. Detailed comments are shown below.

  • The introduction should highlight the role of different group sizes. It is very relevant for this paper. What is the effect of unbalanced clusters on the inference with CRVEs? 
  • I think that CRVE, CR2VE and CR3VE can be better presented. Although it is standard literature, it would be better to explain also the previous proposals with a bit deeper detail. This can help in better highlighting the authors' contribution as well as helping a less expert reader in appreciating more the novelty introduced by the authors;
  • Formula (2): the scaling factor c(n-1)/(v-1)(n-k) is missing. It is important to explain that, although it is asymptotically negligible, it makes CRVE larger when C and N are finite.
  • Section 3, line 70: it would be useful to recall that c is the number of clusters. Further, I would suggest using C (with c be a given cluster c∈C) in referring to the number of clusters. Notation can change accordingly.
  • Section 3, line 78: does λ≥c/(1-c) mean that a greater scaling is needed to ensure unbiasedness of the estimator in presence of unbalanced clusters? Perhaps this point can be discussed more deeply.

Author Response

We are grateful to the referees for their comments, questions and suggestions, from which we
greatly benefited in improving our paper. Below we have copied their texts and have inserted our
reactions in blue.

In this paper, a new type of cluster robust variance estimator (CRVE), that takes into account
the possible difference in groups’ size, is proposed. Moreover, it is unbiased under more general
conditions than the previous CR3VE. The paper contribution is clear and adds new evidence to
the literature. I am happy to say that I enjoyed the reading. However, I have some remarks. To
summarize, the authors should increase their effort in improving the presentation of previous
literature, especially concerning the problem of unequally sized clusters. Detailed comments are
shown below.
The introduction should highlight the role of different group sizes. It is very relevant for
this paper. What is the effect of unbalanced clusters on the inference with CRVEs?

In the Introduction we restrict ourselves to some general remarks that unbalancedness matters but we become quite precise at the end of Section 3. Following on the last point raised by this
referee (please see below) we now say explicitly that taking unbalancedness into account leads to a smaller variance, as it should.

I think that CRVE, CR2VE and CR3VE can be better presented. Although it is standard
literature, it would be better to explain also the previous proposals with a bit deeper detail. This
can help in better highlighting the authors’ contribution as well as helping a less expert reader in appreciating more the novelty introduced by the authors.

We have to say here that we would rather not follow the referee’s suggestion to add detail.
On purpose, we kept Section 2, with the basic theory, as concise as could be. It contains all you
need to know to understand our proposal, with minimal distraction.

Formula (2): the scaling factor c(n − 1)/(v − 1)(n − k) is missing. It is important to explain
that, although it is asymptotically negligible, it makes CRVE larger when C and N are finite.

We called CRVE in (2) “[t]he cluster-robust variance estimator”. This was a bit strong, as
if there was just one such estimator. This is of course not the case. A major variant, e.g. used by
Stata, adds the scaling factor mentioned by the referee (with c for v of course). We have added a
sentence about this.

Section 3, line 70: it would be useful to recall that c is the number of clusters. Further, I
would suggest using C (with c be a given cluster c ∈ C) in referring to the number of clusters.
Notation can change accordingly.

We fully agree. Especially using i to index clusters as we did, not individuals, can be confusing.
We now let C be the number of clusters and index them by c, with c = 1, . . . ,C.

Section 3, line 78: does λ ≥ c/(1 − c) mean that a greater scaling is needed to ensure
unbiasedness of the estimator in presence of unbalanced clusters? Perhaps this point can be
discussed more deeply.

We agree. We show λ ≥ c/(1 − c). The scaling factor is 1/λ, hence with property 1/λ ≤ (1 − c)/c,
the latter being the scaling factor used with CR3VE. So we propose a lower scaling factor and
hence a lower variance to adjust for unbalancedness. This is quite satisfactory since CR3VE tends
to overestimate the variance. We now mention this explicitly in the paper.

Reviewer 4 Report

The paper suggest a modification to the jackknife estimator CR3VE for clustered standard errors with few clusters. The improved version of the CR3VE estimator reduces the bias when estimating standard errors using data with high degress of cluster unbalancedness and high correlation of regressors within clusters.

The paper does not provide any evidence on the practical significance of the results. Hence it is not clear whether resulting improvement in the estimator properties is only cosmetical at best or it is a game-changer to the extent that conclusions on significance of variables of interest obtained with the original CR3VE estimator can be reversed.

In order to see this, a replication of a previous empirical research is necessary with the modified estimator.

Author Response

(The authors gave the same response as above.)

Round 2

Reviewer 3 Report

I am satisfied with the authors' revision.